# Determination of Peak Oxygen Uptake in Patients with Acute Myocardial Infarction: The Role of Arterial Stiffness in Cardio–Vascular–Skeletal Muscle Coupling

**DOI:** 10.3390/jcm13010042

**Published:** 2023-12-21

**Authors:** Akihiro Ogawa, Shinya Kanzaki, Yuki Ikeda, Masahiro Iwakawa, Takahiro Nakagami, Shuji Sato, Hiroshi Mikamo, Satoshi Kido, Arata Nakajima, Kazuhiro Shimizu

**Affiliations:** 1Department of Rehabilitation, Toho University Sakura Medical Center, 564-1, Shimoshizu, Sakura, Chiba 285-8741, Japan; shinya.kanzaki@med.toho-u.ac.jp (S.K.); arata.nakajima@med.toho-u.ac.jp (A.N.); 2Graduate School of Health Sciences, Saitama Prefectural University, 820, Sannomiya, Koshigaya, Saitama 343-8540, Japan; kido-satoshi@spu.ac.jp; 3Department of Internal Medicine, Toho University Sakura Medical Center; 564-1, Shimoshizu, Sakura, Chiba 285-8741, Japan; yuki.ikeda@med.toho-u.ac.jp (Y.I.); masahiro.iwakawa@med.toho-u.ac.jp (M.I.); nak-agami-04@sakura.med.toho-u.ac.jp (T.N.); shuuji.satou@med.toho-u.ac.jp (S.S.); mika-mo-04@sakura.med.toho-u.ac.jp (H.M.); k432@sakura.med.toho-u.ac.jp (K.S.)

**Keywords:** cardio–ankle vascular index, acute myocardial infarction, VO_2 Peak_, cardio–vascular–skeletal muscle coupling

## Abstract

The relationship between arterial stiffness and oxygen uptake (VO_2_) in patients with acute myocardial infarction (AMI) remains unclear. We aimed to investigate this relationship and factors contributing to VO_2_ in patients with AMI. The role of arterial stiffness in cardio–skeletal muscle coupling during exercise was then elucidated. Upon discharge, we measured exercise capacity using cardiopulmonary exercise testing (CPX), assessed arterial stiffness with the cardio–ankle vascular index (CAVI), and determined body composition to assess the skeletal muscle mass of 101 patients with AMI. Patients were categorized based on their CAVI scores into three groups: (i) normal (CAVI: ≤7.9), (ii) borderline (CAVI: 8.0–8.9), and (iii) abnormal (CAVI: ≥9.0). Subsequently, VO_2_ was compared among these groups. The relationship between the CAVI and VO_2 Peak_ during CPX and factors contributing to VO_2 Peak_ were investigated. The abnormal CAVI group had a significantly lower VO_2 Peak_ than the normal and borderline groups. The CAVI was associated with VO_2 Peak_. Furthermore, the CAVI was found to be a factor contributing to VO_2 Peak_. These findings suggest that arterial stiffness in tissue blood distribution and blood supply causes systemic exercise limits in patients with AMI. This suggests that arterial stiffness plays a significant role in cardio–vascular–skeletal muscle coupling.

## 1. Introduction

Aging is a primary factor contributing to the increasing incidence of heart failure (HF) [1]. Several other factors contribute to the increasing prevalence of HF in the older population, including increased life expectancy, accumulated risk factors, and comorbidities. Therefore, addressing the increasing number of older patients with HF is necessary. Additionally, research and development are necessary to elucidate the pathophysiology, prevention, diagnosis, treatment, and rehabilitation of patients with prevalent cardiovascular diseases in Japan [2]. Ischemic heart diseases, such as acute myocardial infarction (AMI), account for approximately 50% of the major etiologies of HF in Japan [3]. The American College of Cardiology Foundation and American Heart Association stages provide useful and complementary information regarding the stages of HF [4]. AMI corresponds to stage B HF, i.e., structural heart disease without signs or symptoms of HF. However, without adequate secondary prevention, HF progresses, leading to physical dysfunction and terminal disease. The decline in physical function, including exercise tolerance, is one of the most important factors associated with the rate of HF exacerbation and mortality in patients with cardiovascular disease (CVD) [5,6,7,8]. 

The onset of AMI is closely intertwined with the progression of atherosclerosis. The aorta’s stiffness serves as a cushion for organ protection and as a windkessel for organ blood flow. This occurs through substantial extension and expansion during cardiac systolic blood ejection [9], effectively facilitating blood supply to major organs. Therefore, dysfunction in arterial stiffness resulting from the advancement of atherosclerosis is linked to heightened organ damage, and especially an increased cardiac burden [10]. Moreover, reduced arterial stiffness is presumed to decrease oxygen delivery to peripheral tissues, thereby impacting blood flow to muscles [11]. Consequently, this can result in a diminished circulatory response during exercise, suboptimal tissue oxygenation, and impaired exercise tolerance. Oxygen uptake (VO_2_) is determined by cardiac and skeletal muscle function and is a common physiological parameter used as an indicator of exercise tolerance and aerobic capacity. In previous studies, higher VO_2max_ values in elderly populations consistently correlated with lower brachial–ankle pulse wave velocity (baPWV) measurements, indirectly highlighting the association between arterial stiffness and cardiorespiratory endurance [12]. However, limitations arise as PWV is contingent on the blood pressure measured at the time of assessment. The cardio–ankle vascular index (CAVI) is acknowledged as a blood pressure-independent marker of arterial stiffness across the entire arterial system, extending from the aortic root to the ankle [13]. Previous studies have reported that an abnormal CAVI value (>9.0) is associated with a significant incidence of cardiovascular events, including cardiovascular death, myocardial infarction, stroke, hospitalization for HF, and coronary revascularization [14]. Therefore, arterial stiffness and VO_2_ are prognostic factors for cardiovascular disease, although the relationship between arterial stiffness and VO_2_ in patients with AMI remains unclear. Blood vessels connect the cardiac and skeletal muscles, and supply oxygen and nutrients throughout the body. They also regulate circulatory dynamics, with arterial stiffness regulating cardiac afterload and blood flow to peripheral tissues [15,16]. Therefore, we defined the response to exercise and physical activity as “the Cardio-vascular-skeletal muscle coupling” and hypothesized that arterial stiffness serves as a determinant of oxygen uptake. This study aimed to investigate the (i) relationship between arterial stiffness and VO_2_, and (ii) the factors contributing to VO_2_ in patients with AMI. This study contributes to the understanding of the role of arterial stiffness in cardio–vascular–skeletal muscle coupling.

## 2. Materials and Methods

### 2.1. Participants

This retrospective observational study used the cardiac rehabilitation database of Toho University Sakura Medical Center. In total, 366 consecutive patients with AMI, who underwent cardiac rehabilitation from admission until discharge between January 2017 and March 2023, were included in this study.

The inclusion criteria were patients who underwent the following: (i) cardiac rehabilitation involving activities such as aerobic exercise, (ii) measurement of arterial stiffness using the CAVI, and (iii) cardiopulmonary exercise testing (CPX). The exclusion criteria were as follows: (i) patients unable to walk independently or perform aerobic exercise, (ii) a lack of CAVI and/or CPX measurements, and (iii) an ankle–brachial index value of ≤0.9. This study included 101 patients (Figure 1). All parameters were evaluated immediately before discharge. 

### 2.2. Ethics

This study was approved by the Ethics Committee of Toho University Sakura Medical Center (ethics number: S22019) and followed the Declaration of Helsinki and the ethical standards of the responsible committee on human experimentation. Patients were given the opportunity to opt out when enrolling in this study. 

### 2.3. Measured Parameters

#### 2.3.1. CAVI

The CAVI is employed in blood pressure-independent assessments of arterial elasticity, serving as a valuable predictor for cardiovascular events, including those triggered by stress and various other factors [17]. The CAVI was assessed using a VaSera1500 vascular screening system (Fukuda Denshi Co., Ltd., Tokyo, Japan) as an indicator, following established protocols [13,16]. The patient was placed in a supine position with their head in a neutral position, and cuffs were applied to both the upper arms and ankles. After a 10 min resting period, measurements were taken to ensure a low cuff pressure of 30–50 mmHg to minimize the cuffs’ impact on hemodynamics. During the examination, blood pressure was recorded, and the CAVI value was calculated using the following equation:CAVI = a {(2p/ΔP) × ln (Ps/Pd) PWV^2^} + b
where Ps and Pd represent systolic and diastolic blood pressure, respectively, PWV indicates the pulse wave velocity from the origin of the aorta to the junction of the tibial and femoral arteries, ΔP is the difference between Ps and Pd, p denotes the blood density, and a and b are constants. 

This equation was derived from Bramwell–Hill’s equation and adjusted for blood pressure based on the stiffness parameter (β). Three skilled examiners carefully observed the pulse waveforms to ensure accurate CAVI assessment in cases of atrial fibrillation.

A CAVI value of ≥8.0 potentially correlates with the presence of asymptomatic or potential atherosclerosis, whereas a CAVI value of ≥9.0 indicates the presence of advanced atherosclerosis and an elevated risk of cardiovascular complications [18]. Consequently, patients in this study were categorized into three groups utilizing their CAVI outcomes as follows: (i) normal (≤7.9), (ii) borderline (8.0–8.9), and (iii) abnormal (≥9.0) [19].

#### 2.3.2. CPX

Cardiorespiratory fitness is a strong predictor of mortality in patients with CVD, with CPX commonly used to measure oxygen uptake as an indicator of cardiorespiratory fitness [7,8]. In this study, CPX and expiratory gas analyses were performed to quantitatively evaluate oxygen uptake (VO_2_). CPX measures VO_2_, ventilatory carbon dioxide output, and respiratory and ventilation rates. CPX also calculates the minute ventilation rate from these measurements [20]. In the CPX test, a Strength Ergo 240 (Mitsubishi Electric Engineering Co., Ltd., Tokyo, Japan) was used for exercise load, an AE-310s Aeromonitor (Minato Medical Science Co., Ltd., Osaka, Japan) was used for breath gas analysis, and an STS-2100 (Nihon Kohden Co., Ltd., Tokyo, Japan) was used for exercise load electrocardiogram. In the study protocol, the patients began with a 3 min warm-up on a bicycle ergometer, followed by a 10 watt/min or 20 watt/min ramp incremental protocol. Continuous 12-lead electrocardiography was performed during the assessment, and blood pressure was recorded every minute during the exercise and recovery periods. After reaching the peak load, all patients pedaled at 0 watts, with a cool-down period of ≥2 min to prevent excessive venous pooling. The testing procedure adhered to published guidelines [21].

#### 2.3.3. Skeletal Muscle Mass and Handgrip Strength

Skeletal muscle mass and handgrip strength tests represent non-invasive, uncomplicated measures commonly utilized for diagnosing sarcopenia [22]. The findings of these measures serve as valuable indicators that can predict rehospitalization rates and prognosis in patients with CVD [23,24]. Skeletal muscle mass was measured using a body composition analyzer (MC-980A; Tanita Corp., Tokyo, Japan) based on direct segmented multifrequency bioelectrical impedance analysis [25]. By inputting the patient’s age, sex, and height, the bioimpedance analysis device yielded segmental muscle mass (arms, legs, and trunk), absolute fat mass, and body fat percentage. The sum of arm and leg skeletal muscle mass represents the appendicular skeletal muscle mass (ASM). The skeletal muscle mass index (SMI) value was defined as the ASM divided by the body height (in meters) squared, as follows: SMI = ASM (kg)/Height (m)^2^

Muscle strength was determined by measuring handgrip strength. An electronic hand dynamometer (Takei Scientific Instruments Co., Ltd., Tokyo, Japan) was used to measure handgrip strength. Two consecutive measurements of handgrip strength were made for both hands and recorded to the nearest kilogram, with the patient standing and the arm of the hand measured parallel to the body.

#### 2.3.4. Echocardiography

Cardiac function was assessed using echocardiography. Left ventricular ejection fraction and diastolic function were assessed according to the American Society of Echocardiography guidelines [26,27]. Tissue Doppler imaging was used to estimate the left ventricular systolic function and early diastolic relaxation by averaging the peak systolic velocity and e′ at the mitral annulus in both the medial and lateral walls. The ratio of early transmitral flow filling velocity (E) to tissue Doppler-derived early diastolic velocity (e′) (E/e′) was employed to estimate left ventricular filling. Trained laboratory technicians conducted all measurements, and a cardiologist verified the results. Echocardiography was performed on admission and before discharge.

#### 2.3.5. Hematology and Biochemistry Data

Data on the peak creatine phosphokinase, albumin (Alb), creatinine, brain natriuretic peptide (BNP), and hemoglobin (Hb) levels were extracted from the clinical records of patients.

### 2.4. Statistical Analyses

Normally distributed data were assessed using the Shapiro–Wilk test. The CAVI values were categorized into three groups (≤7.9, 8.0–8.9, and ≤9.0). Subsequently, the VO_2 Peak_ was compared among these groups using the Kruskal–Wallis test with a Bonferroni adjustment. The statistical significance level was set at *p* < 0.016 (0.05/3 ≈ 0.016).

The relationship between the CAVI value and other clinical parameters was also analyzed using Spearman’s rank correlation coefficient. Finally, the variance inflation factor was examined to assess multicollinearity among the indicators significantly correlated with VO_2 Peak_. Forced-entry multiple regression analysis was used to investigate the VO_2 Peak_ determinants. The results are expressed as the median (interquartile range (IQR)). The statistical significance level was set at 5%, and the analysis was conducted using SPSS Ver29.0 (IBM, Chicago, IL, USA).

## 3. Results

### 3.1. Patient Characteristics

A total of 101 (86 male and 15 female) patients with AMI underwent CPX. The median age was 67.0 (IQR: 54.5, 73.0) and 72.0 (IQR: 68.0, 76.0) years for male and female patients, respectively. The CAVI values were 9.1 (IQR: 8.1, 9.9) and 9.5 (IQR: 8.8, 10.5) for male and female patients, respectively. The patient characteristics are shown in Table 1.

### 3.2. Physical Function and CPX 

The peak VO_2_ (VO_2 Peak_) measured with CPX was 17.9 (IQR: 15.7, 20.9) and 15.4 (IQR: 12.8, 16.3) mL/kg/min for male and female patients, respectively. The physical function and results obtained using CPX are listed in Table 2.

### 3.3. Comparison of VO_2 Peak_ according to CAVI Classification

The VO_2 Peak_ levels in groups classified according to the CAVI values were as follows: (i) 20.1 (IQR: 16.5, 25.2) mL/kg/min in 16 patients with a normal CAVI value (≤7.9), (ii) 18.5 (IQR: 15.9, 22.6) mL/kg/min in 31 patients with a borderline CAVI value (8.0–8.9), and (iii) 16.2 (IQR: 12.8, 18.2) mL/kg/min in 54 patients with an abnormal CAVI value (≥9.0). As shown in Figure 2, patients with an abnormal CAVI value had a significantly lower VO_2 Peak_ than those with normal and borderline CAVI values.

### 3.4. Association between VO_2 Peak_ and Clinical Parameters

Age; the Alb, Hb, and BNP levels; E/e′; the CAVI value; the SMI value; and handgrip strength were significantly correlated with the VO_2 Peak_ in patients with AMI. Table 3 shows the relationship between VO_2 Peak_ and clinical indices, and Figure 3 shows a scatter plot of the CAVI value and VO_2 Peak_ in patients with AMI.

### 3.5. Factors Contributing to VO_2 Peak_

A multiple regression analysis was performed using the forced-entry method to determine the factors contributing to VO_2 Peak_ in patients with AMI. The parameters correlated with VO_2 Peak_ and sex were used as independent variables. The results revealed that handgrip strength and CAVI value significantly contributed to the VO_2 Peak_ in patients with AMI. In comparison, age; sex; Alb, Hb, and BNP levels; E/e′; and SMI values were found to have no significant impact. Table 4 shows the factors identified to contribute to the VO_2 Peak_ in patients with AMI.

## 4. Discussion

This study examined the relationship between the CAVI value and VO_2 Peak_, as well as factors contributing to VO_2 Peak_, to establish a basis for determining the role of arterial stiffness in cardiovascular–skeletal muscle coupling during exercise in patients with AMI.

The study results showed that the CAVI value and VO_2 Peak_ were related (Table 3). Patients with an abnormal CAVI value were found to have a significantly lower VO_2 Peak_ than those with normal and borderline CAVI values (Figure 2). Part of the stroke volume during left ventricular contraction is delivered directly to peripheral tissues, while the other part is stored instantaneously in the aorta and central arteries [28]. The inhibition of arterial dilation during left ventricular ejection due to atherosclerosis limits arterial blood storage and blood supply to peripheral tissues [29]. Shiba et al. [30] reported that with an increase in the CAVI value, the peripheral retinal vascular blood flow changes from steady to pulsatile, and the blood supply decreases. This study hypothesized that a similar reduction in blood flow and supply occurs in the skeletal muscles of patients with AMI. Furthermore, it has been reported that increased cardiac afterload due to atherosclerosis increases myocardial oxygen demand, decreases cardiac output during peak exercise, and decreases cardiorespiratory function [31]. Therefore, the association between the CAVI value and VO_2 Peak_ in this study suggests that arterial stiffness affects the blood supply to skeletal muscles and other organs, resulting in changes in skeletal muscle metabolism during exercise. 

The CAVI value and handgrip strength contributed to the VO_2 Peak_ in patients with AMI. The limiting factors for VO_2_ include the following factors: (i) a reduction in oxygen delivery, (ii) a decrease in cardiac output, and (iii) overperfusion of a small muscle mass during exercise [32]. Therefore, an optimal blood flow distribution to peripheral tissues such as skeletal muscles is important during exercise. During exercise and physical activity, the increased demand for oxygen requires a coordinated response from both the cardiac and skeletal muscles. The cardiovascular system responds by increasing the heart rate, cardiac output, and blood flow to meet the increased metabolic demands of skeletal muscles. Vasodilation of the arteries and redistribution of the blood flow accomplish this response. Thus, the results of this study suggest that impaired arterial stiffness results in decreased blood supply to peripheral tissues and an increased cardiac load, making it difficult to continue exercising and decreasing VO_2_. Additionally, VO_2 Peak_ has been reported to demonstrate an association with CAVI independent of age and visceral fat in hypertensive middle-aged and elderly Japanese men [33]. These findings suggest that oxygen uptake and the CAVI are interconnected factors in hypertensive patients at risk of cardiac disease and in those with AMI. A lower VO_2_ during exercise was associated with a worse long-term prognosis in patients with AMI [34]. Thus, this novel finding in this study suggests that an increased CAVI value also affects the prognosis of patients with AMI. 

Although skeletal muscle mass is a determinant of VO_2_ [35], the SMI, an index of skeletal muscle mass calculated from body composition, did not contribute to VO_2 Peak_ in this study. However, handgrip strength was identified as a contributing factor. The Health, Aging, and Body Composition Study [36] reported that mortality risk was strongly associated with quadricep power and handgrip strength but not with muscle mass. The study also reported that measures of muscle quality were more important in estimating mortality risk. The results of this study also support the importance of skeletal muscle quality, such as muscle performance. Furthermore, handgrip strength (HGS) has been reported to correlate with exercise capacity, as determined by the 6-minute walk distance (6MWD), in patients with coronary heart disease. This can be used as a predictive index for determining the degree of exercise capacity [37]. Both 6MWD and oxygen uptake serve as indicators of exercise tolerance, suggesting that muscle strength, including grip strength, contributes to exercise tolerance. At the same time, it is possible that qualitative changes in skeletal muscle, which may induce a decline in muscle strength, may also impact patients with AMI. Skeletal muscle dysfunction accompanying HF is not limited to skeletal muscle atrophy but also involves muscle fiber type transformation, leading to decreased exercise tolerance [38]. This indicates that the metabolic function of skeletal muscle, rather than cardiac function, is an important determinant of oxygen consumption [39]. Patients with AMI in this study were likely to have skeletal muscle dysfunction similar to that in those with HF. Therefore, skeletal muscle mass measurements alone do not adequately capture skeletal muscle dysfunction in patients with cardiac disease.

In summary, this study obtained a novel insight regarding the role of the CAVI in relation to VO_2 Peak_. It was suggested that arterial stiffness determines blood supply from the arteries and its distribution to peripheral tissues through cardiac afterload. Furthermore, blood flow distribution to skeletal muscles contributes to energy metabolism, and it was speculated that skeletal muscle function plays an important role in VO_2_.

This study had some limitations. First, the number of patients with AMI who underwent CPX was low (27.6%). This was because some patients could not undergo CPX due to their decline in physical function and old age. Notably, patients with AMI who did not undergo CPX tended to be older than those who did, with a median age of 72.0 years. Moreover, the proportion of females was significantly lower, accounting for only 14.9% of the total sample size. Therefore, this study may have been limited to patients with AMI and relatively preserved motor function who could undergo CPX. On the other hand, the authors have previously reported an association between the 6 min walking distance and CAVI value in older patients with HF aged ≥65 years [40]. Together with the present results, this suggests an association between exercise tolerance and arterial stiffness in patients with heart disease. Second, cautious clinical assessment was considered necessary to determine skeletal muscle mass using bioelectrical impedance analysis. Skeletal muscle tissue generally contains much more water than fat tissue. Although bioelectrical impedance analysis uses this property to measure the body composition, edema may overestimate skeletal muscle tissue in some patients with AMI. However, the patients with AMI in this study received cardiac rehabilitation during their hospitalization, and their body composition was measured at discharge when their condition was stable. Therefore, edema was considered to have little effect on body composition.

## 5. Conclusions

This study showed that VO_2 Peak_ and CAVI values were associated at discharge in patients with AMI, and that handgrip strength and CAVI value contributed to VO_2 Peak_. This study suggested that arterial stiffness plays a role during whole-body endurance exercise in cardiovascular–skeletal muscle coupling. This indicates the importance of measuring arterial stiffness as an index of cardiac rehabilitation in patients with AMI.

## Figures and Tables

**Figure 1 jcm-13-00042-f001:**
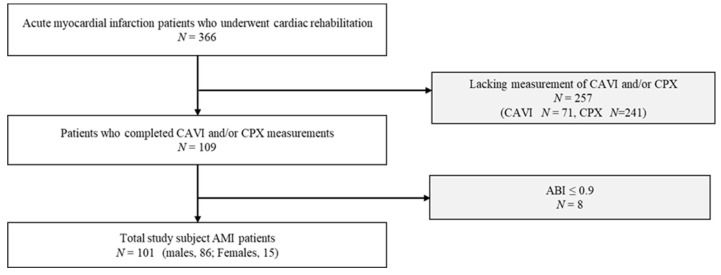
Study enrolment procedure. Abbreviations: CAVI, cardio–ankle vascular index; ABI, ankle–brachial index.

**Figure 2 jcm-13-00042-f002:**
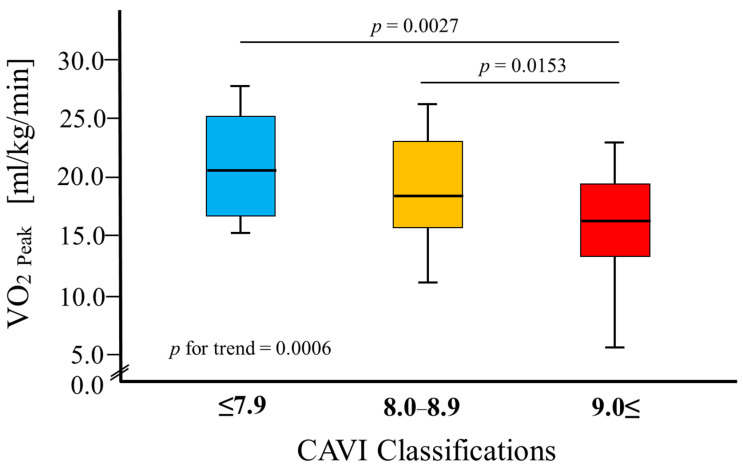
Comparison of peak oxygen uptake according to cardio–ankle vascular index classification. Abbreviation: VO_2_, oxygen uptake.

**Figure 3 jcm-13-00042-f003:**
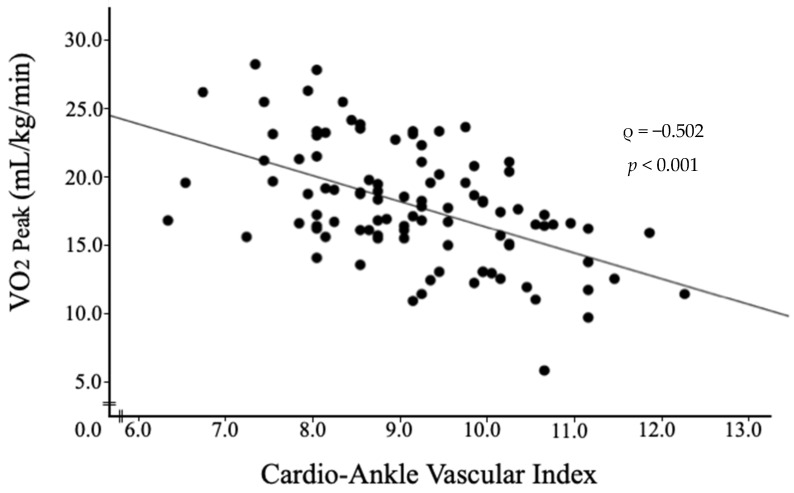
Scatter plot of the cardio–ankle vascular index and peak oxygen uptake in patients with acute myocardial infarction.

**Table 1 jcm-13-00042-t001:** Patient characteristics.

Variable	Patients with AMI *n* = 101
Male, *n* (%)	86 (85.1)
Age, years	67.5 (56.0, 74.0)
BMI, kg/m^2^	22.3 (20.9, 25.2)
sBP, mmHg	115.0 (106.5, 126.5)
dBP, mmHg	71.0 (63.5, 80.5)
HR, bpm	68.0 (59.3, 77.8)
Alb, g/dL	3.8 (3.6, 4.2)
Cre, mg/dL	0.92 (0.82, 1.06)
Hb, mg/dL	13.4 (11.9, 14.2)
CPK _Peak_	1162.0 (374.5, 2240.0)
BNP, pg/mL	86.9 (52.0, 163.2)
EF, %	61.0 (53.5, 67.0)
E/e′	11.2 (9.0, 15.5)
CAVI	9.1 (8.2, 9.9)
Smoking, *n* (%)	34 (33.7)
Complications, *n* (%)	
AF	5 (4.9)
HT	42 (41.6)
DM	31 (31.7)
DL	63 (62.4)
Use of medications, *n* (%)	
Ca-antagonists	9 (8.9)
RAS-inhibitors	53 (52.5)
β-blockers	72 (71.3)
Diuretics	28 (27.7)
Statins	76 (75.2)
Nitrate drug	42 (41.6)

Abbreviations: AMI, acute myocardial infarction; BMI, body mass index; sBP, systolic blood pressure; dBP, diastolic blood pressure; HR, heart rate; Alb, albumin; Cre, creatinine; CPK, creatine phosphokinase; BNP, brain natriuretic peptide; EF, ejection fraction; E/e′, ratio of early transmitral filling velocity to early diastolic velocity; CAVI, cardio–ankle vascular index; AF, atrial fibrillation; HT, hypertension; DM, diabetes mellitus; DL, dyslipidemia; Ca-antagonists, calcium antagonists; RAS-inhibitors, renin–angiotensin–aldosterone inhibitors. Data are presented as medians (interquartile ranges), unless otherwise indicated.

**Table 2 jcm-13-00042-t002:** Assessment of physical parameters and cardiopulmonary exercise testing.

Variable	Patients with AMI *n* = 101
SMI, kg/m^2^	6.37 (5.68, 7.38)
Handgrip strength, kg	21.7 (16.2, 28.2)
VO_2 AT_, mL/kg/min	13.7 (11.4, 15.7)
VO_2 Peak_, mL/kg/min	17.7 (15.6, 20.7)
HR _AT_, beat/min	104.5 (94.8, 111.0)
HR _Peak_, beat/min	124.5 (110.3, 134.0)
VO_2_/HR _Peak_, mL/beat	9.5 (7.9, 10.9)
VE vs. VCO_2_ slope	32.0 (29.1, 37.0)
ΔVO_2_/ΔWR, mL/min/watt	8.1 (7.0, 9.5)

Abbreviations: SMI, skeletal muscle mass index; VO_2_, oxygen uptake; AT, anaerobic threshold; VE, ventilation rate; VCO_2_, ventilatory carbon dioxide output; WR, work rate. Data are presented as medians (interquartile ranges).

**Table 3 jcm-13-00042-t003:** Correlation between peak oxygen uptake and clinical parameters.

Variable	Correlation Coefficient (ρ)	95% CI	*p*-Value
Age	−0.487	−0.628–−0.315	<0.001
BMI	0.136	−0.075–0.335	0.191
sBP	−0.026	−0.230–0.180	0.798
dBP	0.024	−0.182–0.228	0.812
CPK_Peak_	−0.09	−0.410–0.251	0.598
Alb	0.297	0.094–0.476	0.004
Cre	−0.121	−0.319–0.087	0.240
Hb	0.454	0.270–0.603	<0.001
BNP	−0.326	−0.509–−0.115	0.002
EF	0.049	−0.204–0.296	0.698
E/e′	−0.212	−0.403–−0.004	0.04
CAVI	−0.502	−0.640–−0.333	<0.001
SMI	0.479	0.302–0.624	<0.001
Handgrip strength	0.501	0.325–0.643	<0.001

Abbreviations: Hb, hemoglobin. Spearman’s rank correlation coefficient was analyzed. The significance level was set at *p* < 0.05.

**Table 4 jcm-13-00042-t004:** Factors which contributed to the peak oxygen uptake in patients with acute myocardial infarction.

Variable	*r*	*r* ^2^	β	95% CI	*p*-Value
Age	0.657	0.432	0.098	–4.494–35.936	0.125
Sex			−0.103	–4.132–1.917	0.467
Alb			−0.011	–2.243–2.042	0.926
Hb			0.106	–0.296–0.789	0.368
BNP			−0.046	–0.008–0.005	0.678
E/e′			−0.043	–0.225–0.149	0.684
CAVI			−0.258	–1.856–−0.070	0.035
SMI			−0.014	–1.170–1.075	0.933
Handgrip strength			0.541	0.074–0.402	0.005

A forced-entry multiple regression analysis was used to investigate VO_2 peak_ determinants. Adjustment factors: age; sex; Alb, Hb, and BNP; E/e′; CAVI; SMI; and handgrip strength. Abbreviations: R, multiple correlation coefficient; R^2^, coefficient of determination; β, standardized partial regression coefficient; CI, confidence interval.

## Data Availability

The original contributions presented in this study are included in the article, and further inquiries can be directed to the corresponding author.

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
