# Peer review of "Determination of Peak Oxygen Uptake in Patients with Acute Myocardial Infarction: The Role of Arterial Stiffness in Cardio–Vascular–Skeletal Muscle Coupling"

_jcm, 2023, doi:10.3390/jcm13010042_

Round 1

Reviewer 1 Report

Comments and Suggestions for Authors

This study fills research gaps and contributes to our understanding of the role of arterial stiffness in the coupling of cardiovascular and skeletal muscle.

In general, the manuscript is well written, coherent, and follows a logical development with good spelling and grammar, as well as a good scientific style.
In relation to the summary allows to understand the article without having to read it. The introduction and method fit the standards perfectly.
In relation to the results, they are fine
The discussion follows the same logical order as the results and also includes the limitations.
In relation to the conclusions, it is fine.

Minor comments: 8 bibliographic items out of 29 are from before 2010. It is worth adding a few recent bibliographic items in the introduction or discussion part

It is worth emphasizing the strengths of the study and writing about the gaps it fills

Author Response

Reviewer 1

Thank you very much for your valuable comments on our manuscript. The following is our response to your comments. Corrections in the text are noted in red.

This study fills research gaps and contributes to our understanding of the role of arterial stiffness in the coupling of cardiovascular and skeletal muscle. In general, the manuscript is well written, coherent, and follows a logical development with good spelling and grammar, as well as a good scientific style. In relation to the summary allows to understand the article without having to read it. The introduction and method fit the standards perfectly.

In relation to the results, they are fine. The discussion follows the same logical order as the results and also includes the limitations. In relation to the conclusions, it is fine.

Minor comments: 8 bibliographic items out of 29 are from before 2010. It is worth adding a few recent bibliographic items in the introduction or discussion part It is worth emphasizing the strengths of the study and writing about the gaps it fills

Response:

Thank you for your valuable comments.

Older references have been cited, and references to the original works, particularly regarding methodology, have been included. However, as you have mentioned, it is equally important to explore more recent literature. Accordingly, the introduction, methods, and discussion sections have been revised with the incorporation of recent references. The additional references include reference numbers 7, 8, 9, 10, 11, 12, 17, 22, 23, 24, 33, and 37.

Reviewer 2 Report

Comments and Suggestions for Authors

Determination of Peak Oxygen Uptake in Patients with Acute Myocardial Infarction: The Role of Arterial Stiffness in the 3 Cardio-Vascular-Skeletal Muscle Coupling

I am grateful for the opportunity to review this innovative work, which contributes significantly to our understanding of acute myocardial infarction (AMI) and its physiological implications. The study offers valuable insights into the relationship between arterial stiffness, VO2 Peak, and exercise tolerance in AMI patients. However, some minor modifications as per the suggestions outlined below could further enhance the impact and clarity of the research

Abstract

The abstract is informative and covers the essential aspects of the study. Minor modifications for clarity, accessibility, and emphasis on the study's implications could further enhance its effectiveness. Similarly, the study's title should be concise yet descriptive, capturing these critical elements while remaining engaging for potential readers.

Introduction

The introduction effectively highlights the increasing prevalence of heart failure (HF) in aging populations, particularly emphasizing its relevance in Japan. It touches upon the significant role of ischemic heart diseases like acute myocardial infarction (AMI) in the etiology of HF. However, the link between AMI, arterial stiffness, and oxygen uptake (VO2) could be clarified further. While rich in technical detail, including explanations of VO2 and the cardio-ankle vascular index (CAVI), the introduction would benefit from simplification for broader accessibility. The study's aims and hypotheses are clearly stated, focusing on exploring the relationship between arterial stiffness and VO2 in AMI patients.

Methods and Statistical Analysis

The retrospective observational design selected for this study is appropriate for its aims. However, it would be beneficial for the authors to offer more insights into the choice of the specific patient database used. Discussing how representative this sample is of the broader acute myocardial infarction (AMI) patient population would greatly aid in understanding the potential generalizability of the study's findings.

While the inclusion and exclusion criteria are clearly outlined, the manuscript would benefit from a more comprehensive rationale. Detailed explanations of why specific criteria, particularly CAVI measurements and CPX testing, were chosen would provide valuable context. This would enhance the study by clarifying the relevance of these criteria in assessing AMI patients.

In terms of measurement techniques, the methods for assessing arterial stiffness, cardiopulmonary exercise testing, and other parameters are described in detail. To strengthen this section further, the authors should consider elaborating on the rationale behind selecting these specific techniques. Explaining their relevance and significance in relation to the study's objectives would provide a deeper understanding of the methodological choice.

Results

The presentation of results in the study is well-executed. The patient characteristics section efficiently outlines the demographics of the 101 AMI patients who underwent CPX, providing median ages and CAVI values for both male and female participants. These details are neatly summarized in Table 1. Furthermore, the physical function and CPX results are concisely presented, with VO2 Peak measurements for both genders and Table 2 offers a comprehensive overview of physical parameters and CPX findings.

The comparison of VO2 Peak according to CAVI classification is delineated, showing significant differences in VO2 Peak across the categorized CAVI values. Additionally, the association between VO2 Peak and clinical parameters is effectively demonstrated through correlation analysis, with a helpful visual representation in Figure 3. Lastly, the multiple regression analysis in section 3.5 insightfully identifies factors contributing to VO2 Peak in patients with AMI, with these results detailed in Table 4. The overall structuring and clarity in the presentation of these results are commendable.

Discussion

In the Discussion section, the study provides valuable insights into the interplay between CAVI value, VO2 Peak, and factors influencing VO2 Peak in AMI patients, mainly focusing on arterial stiffness's role in exercise physiology. It finds a significant correlation between abnormal CAVI values and lower VO2 Peaks. This finding resonates with existing literature on arterial blood storage and supply limitations due to atherosclerosis. This study's hypothesis, drawing parallels between reduced blood flow in skeletal muscles of AMI patients and peripheral retinal vascular changes, is particularly noteworthy.

Interestingly, the study links CAVI value and handgrip strength with VO2 Peak, shedding light on factors like oxygen delivery reduction and cardiac output decrease. The implications of impaired arterial stiffness leading to decreased blood supply and increased cardiac load during exercise are significant. Notably, while skeletal muscle mass is typically a determinant of VO2, this study finds that the Skeletal Muscle Mass Index (SMI) does not contribute to VO2 Peak, in contrast to handgrip strength. This finding aligns with existing research emphasizing muscle quality over muscle mass in estimating mortality risk, highlighting the importance of muscle quality in AMI patients.

However, the results would benefit from a comparison with other studies, particularly in analysing variables like VO2 Peak, CAVI values, and handgrip strength, to contextualize these findings within the broader field. Such comparisons could provide a more comprehensive understanding of how these variables interact in different patient populations or under varying conditions.

Conclusion

The conclusion effectively highlights the significant association between VO2 Peak, CAVI value, and handgrip strength in AMI patients, underlining the crucial role of arterial stiffness in cardiac rehabilitation.

References

Verifying that the references comply with the journal's guidelines is essential. Additionally, the manuscript would benefit from including recent studies on arterial stiffness, VO2 Peak, and cardiovascular rehabilitation in AMI patients. This will ensure the reference list is up-to-date and broaden the context of the study's findings, enhancing its relevance and depth in the field.

Author Response

Reviewer 2

Thank you very much for your valuable comments on our manuscript. The following is our response to your comments. Corrections in the text are noted in red.

Determination of Peak Oxygen Uptake in Patients with Acute Myocardial Infarction: The Role of Arterial Stiffness in the 3 Cardio-Vascular-Skeletal Muscle Coupling

I am grateful for the opportunity to review this innovative work, which contributes significantly to our understanding of acute myocardial infarction (AMI) and its physiological implications. The study offers valuable insights into the relationship between arterial stiffness, VO2 Peak, and exercise tolerance in AMI patients. However, some minor modifications as per the suggestions outlined below could further enhance the impact and clarity of the research

Response:

Thank you for your valuable comments.

The title was selected to underscore the role of blood vessels connecting the heart and skeletal muscles in the context of exercise.

Abstract

The abstract is informative and covers the essential aspects of the study. Minor modifications for clarity, accessibility, and emphasis on the study's implications could further enhance its effectiveness. Similarly, the study's title should be concise yet descriptive, capturing these critical elements while remaining engaging for potential readers.

Introduction

The introduction effectively highlights the increasing prevalence of heart failure (HF) in aging populations, particularly emphasizing its relevance in Japan. It touches upon the significant role of ischemic heart diseases like acute myocardial infarction (AMI) in the etiology of HF. However, the link between AMI, arterial stiffness, and oxygen uptake (VO2) could be clarified further. While rich in technical detail, including explanations of VO2 and the cardio-ankle vascular index (CAVI), the introduction would benefit from simplification for broader accessibility. The study's aims and hypotheses are clearly stated, focusing on exploring the relationship between arterial stiffness and VO2 in AMI patients.

Response:

Thank you for your valuable comments.

I have incorporated an explanation of the progression of acute myocardial infarction resulting from arterial elasticity disorders and the development of atherosclerosis. Additionally, I have cited previous studies addressing reduced oxygen uptake.

Correction point: lines 49–58, lines 60–66, lines 73–74 (page 2).

Methods and Statistical Analysis

The retrospective observational design selected for this study is appropriate for its aims. However, it would be beneficial for the authors to offer more insights into the choice of the specific patient database used. Discussing how representative this sample is of the broader acute myocardial infarction (AMI) patient population would greatly aid in understanding the potential generalizability of the study's findings.

While the inclusion and exclusion criteria are clearly outlined, the manuscript would benefit from a more comprehensive rationale. Detailed explanations of why specific criteria, particularly CAVI measurements and CPX testing, were chosen would provide valuable context. This would enhance the study by clarifying the relevance of these criteria in assessing AMI patients.

Response:

Details of the inclusion and exclusion criteria have been included in the text and Figure 1. Additionally, we have provided characteristics of patients who were unable to undergo cardiopulmonary exercise testing (CPX) within the study constraints, to broaden the generalizability of this study to patients.

Correction point: Lines 83–85 (page 2), lines 87–90 (page 2), lines 90–92 (page 2), Figure 1, lines 320-323 (page 10).

In terms of measurement techniques, the methods for assessing arterial stiffness, cardiopulmonary exercise testing, and other parameters are described in detail. To strengthen this section further, the authors should consider elaborating on the rationale behind selecting these specific techniques. Explaining their relevance and significance in relation to the study’s objectives would provide a deeper understanding of the methodological choice.

Response:

Thank you for your valuable comments. Regarding the selection criteria for the measurement items in this study, we have supported the rationale by incorporating the usefulness of these items by referring to previous studies, thereby, providing additional support for their inclusion in this investigation.

Correction point: Lines 102–104 (page 3), lines 125–128 (page 3), lines 142–145 (page 4).

Results

The presentation of results in the study is well-executed. The patient characteristics section efficiently outlines the demographics of the 101 AMI patients who underwent CPX, providing median ages and CAVI values for both male and female participants. These details are neatly summarized in Table 1. Furthermore, the physical function and CPX results are concisely presented, with VO2 Peak measurements for both genders and Table 2 offers a comprehensive overview of physical parameters and CPX findings.

The comparison of VO2 Peak according to CAVI classification Is delineated, showing significant differences in VO2 Peak across the categorized CAVI values. Additionally, the association between VO2 Peak and clinical parameters is effectively demonstrated through correlation analysis, with a helpful visual representation in Figure 3. Lastly, the multiple regression analysis in section 3.5 insightfully identifies factors contributing to VO2 Peak in patients with AMI, with these results detailed in Table 4. The overall structuring and clarity in the presentation of these results are commendable.

Response:

We are deeply honored to receive such positive comments. Our aspiration is that the outcomes of this study prove valuable to a broad readership.

Discussion

In the Discussion section, the study provides valuable insights into the interplay between CAVI value, VO2 Peak, and factors influencing VO2 Peak in AMI patients, mainly focusing on arterial stiffnes’'s role in exercise physiology. It finds a significant correlation between abnormal CAVI values and lower VO2 Peaks. This finding resonates with existing literature on arterial blood storage and supply limitations due to atherosclerosis. This stud’'s hypothesis, drawing parallels between reduced blood flow in skeletal muscles of AMI patients and peripheral retinal vascular changes, is particularly noteworthy.

Interestingly, the study links CAVI value and handgrip strength with VO2 Peak, shedding light on factors like oxygen delivery reduction and cardiac output decrease. The implications of impaired arterial stiffness leading to decreased blood supply and increased cardiac load during exercise are significant. Notably, while skeletal muscle mass is typically a determinant of VO2, this study finds that the Skeletal Muscle Mass Index (SMI) does not contribute to VO2 Peak, in contrast to handgrip strength. This finding aligns with existing research emphasizing muscle quality over muscle mass in estimating mortality risk, highlighting the importance of muscle quality in AMI patients.

However, the results would benefit from a comparison with other studies, particularly in analysing variables like VO2 Peak, CAVI values, and handgrip strength, to contextualize these findings within the broader field. Such comparisons could provide a more comprehensive understanding of how these variables interact in different patient populations or under varying conditions.

Response:

Thank you for your valuable feedback. In response, we have expanded the discussion on VO2, CAVI, and handgrip strength by incorporating insights from previous studies, exploring their relationships and indirect effects.

Correction point: Lines 284–288 (page 9), lines 298–303 (page 10).

Conclusion

The conclusion effectively highlights the significant association between VO2 Peak, CAVI value, and handgrip strength in AMI patients, underlining the crucial role of arterial stiffness in cardiac rehabilitation

Response:

I am sincerely grateful for your positive feedback. I hope that the findings of this study prove to be valuable to a wide audience of readers.

References

Verifying that the references comply with the journal's guidelines is essential. Additionally, the manuscript would benefit from including recent studies on arterial stiffness, VO2 Peak, and cardiovascular rehabilitation in AMI patients. This will ensure the reference list is up-to-date and broaden the context of the study's findings, enhancing its relevance and depth in the field.

Response:

As a result of the revisions and additions made in response to the reviewers’ comments, we believe that the literature now incorporates significantly more recent studies, thereby further enhancing the value of this paper.

 The additional references include reference numbers 7, 8, 9, 10, 11, 12, 17, 22, 23, 24, 33, and 37.
